**Brief Communication**

# Vitessce: integrative visualization of multimodal and spatially resolved single-cell data

**Mark S. Keller** [1], **Ilan Gold** [1], **Chuck McCallum**[1], **Trevor Manz** [1], **Peter V. Kharchenko** [1,2,3] **& Nils Gehlenborg** [1] ✉

Multiomics technologies with single-cell and spatial resolution make it possible to measure thousands of features across millions of cells. However, visual analysis of high-dimensional transcriptomic, proteomic, genome-mapped and imaging data types simultaneously remains a challenge. Here we describe Vitessce, an interactive web-based visualization framework for exploration of multimodal and spatially resolved single-cell data. We demonstrate integrative visualization of millions of data points, including cell-type annotations, gene expression quantities, spatially resolved transcripts and cell segmentations, across multiple coordinated views. The open-source software is available at http://vitessce.io.

Techniques for characterizing tissues and organs at single-cell resolution have recently overcome the challenges of spatial localization and measurement of multiple modalities, improving our ability to distinguish cell types, cell states and cell neighborhoods[1]. The high dimensionality of single-cell omics and imaging data has catalyzed the development of computational models designed to annotate cellular observations with segmentations, cluster memberships and predicted cell types. Visual analysis of multimodal and spatially resolved single-cell datasets which integrate genomics, transcriptomics and imaging modalities can facilitate quality control, communication of results, identification of biomarkers and generation of hypotheses.

Despite experimental and computational advances in the field, there remains a lack of scalable tools for interactive visualization of multimodal and spatial single-cell datasets. The lack of sufficient tools for visual analysis of heterogeneous biological datasets, termed 'the challenge of ultrascale integrative visualomics', has been highlighted by the visualization research community[2,3]. Visualization of multimodal single-cell datasets currently requires multiple disconnected tools such as image viewers and genome browsers which support individual data modalities. Tools that facilitate discovery of connections across modalities will enable multimodal experimental data to be used to their full potential to improve understanding of health, disease and regulatory mechanisms[4].

Recent tools for visualization of single-cell data have focused on interactivity and scalability. Tools such as Cellxgene[5], Cirrocumulus[6] and Pagoda2[7] support interactive exploration of datasets containing millions of cells and thousands of genes using scatterplots and heatmaps. However, these tools are designed for visual analysis of transcriptomics data. As a result, they lack support for spatial data types such as cell segmentations or genome-mapped modalities such as chromatin accessibility profiles. On the other hand, tools designed for spatial data such as TissUUmaps[8] and Napari-SpatialData[9] do not support nonspatial or genome-mapped data visualization. The focus so far on single-modality data has resulted in tools that do not support visualization tasks such as comparison and relation across modalities and in spatial context[10].

We developed Vitessce to overcome these limitations and enable integrative visualization of multimodal and spatial single-cell data. Vitessce is a scalable web-based data visualization framework that supports simultaneous visual exploration of transcriptomics, proteomics, genome-mapped and imaging modalities (Fig. 1). For example, the software provides modules based on Viv[11] and HiGlass[12] toolkits for visualization of multiscale multiplexed imaging data and genome-mapped data, respectively (Fig. 2). A strength of Vitessce that distinguishes it from prior work is the ability to deploy visualizations of spatial and

[1]Department of Biomedical Informatics, Harvard Medical School, Boston, MA, USA. [2]Broad Institute of MIT and Harvard, Cambridge, MA, USA. [3]Present address: Altos Labs, San Diego, CA, USA. ✉e-mail: nils@hms.harvard.edu

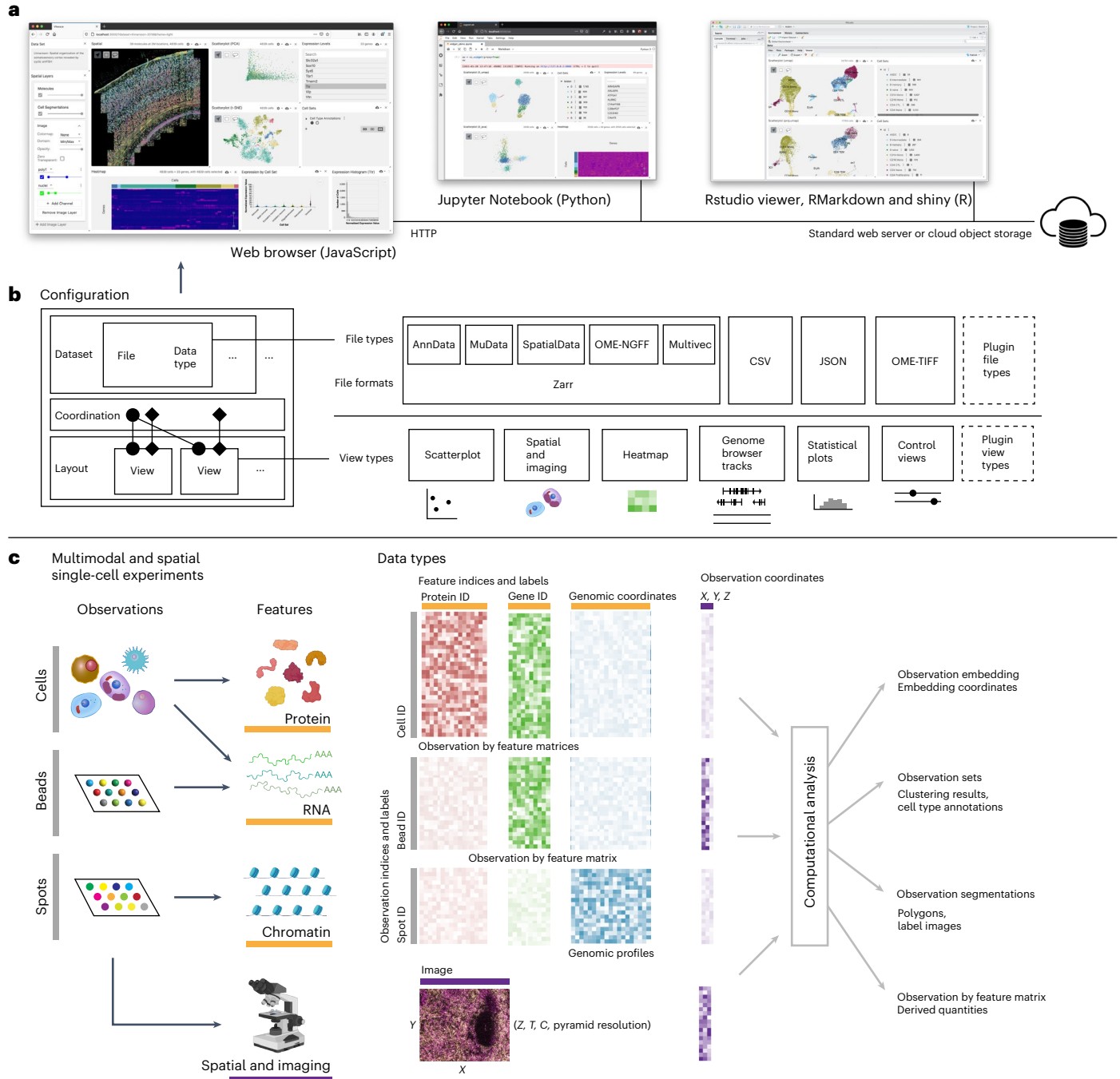

**Fig. 1 | Vitessce can be used in multiple settings and can be configured to visualize raw and derived measurements from multimodal and spatial single-cell experiments. a**, Vitessce can be used as a JavaScript component in a web browser or a widget in Python and R analysis environments. **b**, Single-cell, single-molecule and microscopy data stored in multiple formats can be visualized in multiple types of views (that is, interactive visualizations). **c**, The modular design of Vitessce enables integrative visualization of multimodal and spatial single-cell experiments alongside computational analysis results. The arrows between observations and features represent the ability to visualize data from heterogeneous experiments that measure subsets of features that are shared by subsets of observations. OME, open microscopy environment; NGFF, next generation file format; CSV, comma separated values; JSON, JavaScript object notation; TIFF, tagged image file format; ID, identifier. Icon credits: cell images under 'Observations' from Reactome.org under a Creative Commons license CC BY 4.0; images under 'Features' from https://www.biorender.com/.

single-cell data in a wide range of contexts and compute infrastructures, including in static websites, web applications, data portals, Jupyter Notebooks, RStudio and R Shiny apps.

Data from technologies such as single-cell RNA sequencing (scRNA-seq)[13] and single-cell assay for transposase-accessible chromatin by sequencing (scATAC-seq)[14], as well as multimodal and spatial technologies such as cellular indexing of transcriptomes and epitopes by sequencing (CITE-seq)[15], co-detection by indexing (CODEX)[16], Slide-seq[17], 10x Genomics Visium and Multiome can be visualized with Vitessce. Data (pre)processing (for example, dimensionality reduction and cell segmentation) must be performed outside of Vitessce or through plugins[18], decoupling visualizations from bioinformatic algorithms. Vitessce can access files stored on static web servers and cloud object storage, supporting AnnData[19,20], MuData[21], SpatialData[9],

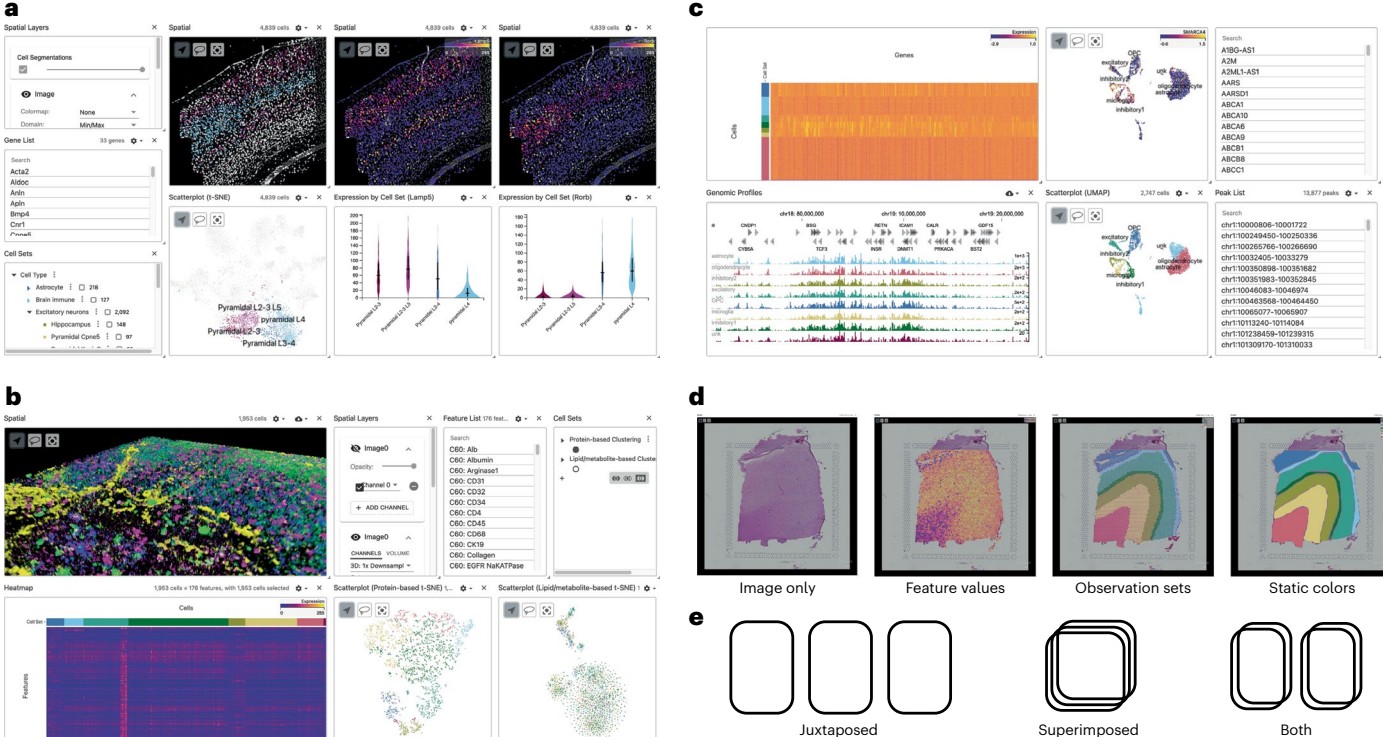

**Fig. 2 | Sample use cases for Vitessce. a**, Visualization of a single-molecule FISH experiment containing multiplexed images, spatially resolved RNA molecules and cell segmentations[37]. **b**, Volumetric rendering of a three-dimensional multimodal imaging mass spectrometry (IMS) dataset alongside a heatmap and side-by-side protein-based and lipid/metabolite-based dimensionality reduction scatterplots[40]. **c**, Simultaneous visualization of gene expression and chromatin accessibility from a 10x Genomics Multiome dataset. **d**, Schematic

of supported visualization layers in the spatial view of Vitessce, namely, points, spots, segmentations and images[41]. Spot and segmentation layers can be colored by feature values, set membership or static colors. **e**, Vitessce supports both juxtaposed and superimposed arrangements of multiple spatially resolved visualizations. Chr, chromosome; t-SNE, t-distributed stochastic neighbor embedding; UMAP, uniform manifold approximation and projection.

comma separated values (CSV), open microscopy environment-tagged image file format (OME-TIFF)[22,23] and OME-Zarr[24,25] formats.

Given the rapidly expanding application of single-cell and spatial assays, the Vitessce framework is designed to reach a large audience of visualization producers and consumers. This audience includes single-cell data analysts intending to communicate results with collaborators, readers of publications who are interested in questions beyond those answered by static figures and consumers of single-cell databases interested in checking data quality before follow-up studies. The framework addresses five key challenges to enable a wide audience to construct, deploy and access visualizations of multimodal and spatially resolved single-cell data:

1. **Tailor visualizations to problem-specific data and biological questions**: Depending on the experimental approach, a single-cell dataset may contain one or more modalities with or without spatial coordinates and images (Fig. 1c). Downstream analysis approaches vary widely and result in derived data types such as dimensionality reductions, clusterings, cell-type annotations and cell segmentations[26]. Heterogeneity in data types and in the types of questions asked by single-cell biologists require different visual representations and interactions, for example, moving between expression and physical space. For multimodal datasets, analysts can use Vitessce to visualize modalities independently (that is, different modalities in different plots) or to visualize the precomputed results of an integration method that projects multiple modalities into a shared low-dimensional space.

    Vitessce enables visual analysis of a patchwork of data types and supports multiple types of questions by providing a collection

of interactive views that can be linked and coordinated using a modular approach. These views enable visualization of datasets containing highly multiplexed images, cell and organelle segmentations, cell-by-gene expression matrices, cell-by-peak count matrices, cluster-level chromatin accessibility profiles, flat and hierarchical cell-type annotations, and dimensionality reductions. Multiple views can be arranged in a grid layout that allows simultaneous visualization of data from multiple experiments. The views in Vitessce are designed to scale to large datasets. For example, the scatterplots can display millions of cells and the heatmaps can display tens of thousands of features (for example, genes, proteins or chromatin accessibility peaks) by using WebGL and deck.gl[27]. Viv[11], HiGlass[12] and Vega-Lite[28] are used to support visualization of multigigabyte images, genome-wide measurements and basic statistical analyses, respectively. We provide documentation to facilitate development of additional view types as plugins (Supplementary Note 1).

2. **Integrate and explore multimodal data with coordinated multiple views**: Visual analysis of multimodal and spatial single-cell data involves identification and relation of patterns that span modalities and data types. The current ecosystem of disparate tools such as genome, scatterplot and image browsers makes such tasks difficult. Vitessce addresses the challenge of relation across modalities by supporting coordinated multiple views[29], enabling interactions such as selection of genes and cell types to be reflected in multiple visualizations (Fig. 1b). Visualization producers can configure any subset of views to be linked on any subset of over 50 parameters, including visualization properties and user preferences. Changes in coordinated values

are reflected in configuration updates, enabling snapshots of state to be captured. Configurations can be saved and shared via uniform resource locator (URL), making the interactive visualization reproducible and trackable[30].

3. **Explore visualizations in different computational environments**: Visual inspection of data is crucial to single-cell data analysis[31]. Vitessce can be integrated directly into not only web applications, but also exploratory computing environments where data analysis is performed. In Jupyter Notebooks and RMarkdown, Vitessce can be used as an interactive widget to visualize analysis results alongside the corresponding code (Fig. 1b). Vitessce R and Python widgets are accompanied by object-oriented application programming interfaces (APIs) to simplify configuration. Widgets can be configured with local or remote data, enabling visualization of datasets that may be located in the cloud or on a compute cluster. These APIs are compatible with the data structures used by popular single-cell data analysis packages, including Scanpy[19,32], SnapATAC[33] and Seurat[34]. Export functions are included in the APIs to simplify remote data hosting.

4. **Deploy and share interactive visualizations**: Interactive visualizations can be more dynamic, scalable, shareable and easier to deploy when they do not rely on a specialized server, as Manz et al.[11] demonstrate for multiplexed bioimaging visualizations. Implemented as a client-side web application without server-side requirements, Vitessce loads data from static files stored on standard web servers or cloud object storage systems including Google Cloud Storage and Amazon Web Services S3. Data loading, transformation and rendering processes in Vitessce are performed directly in the web browser. While the JavaScript framework itself does not rely on a specialized server, integrations with server-side computing infrastructure can be achieved through the development of plugin views which rely on such infrastructure internally. This is demonstrated by Cheng et al.[18] in plugin views that communicate with a specialized server for data integration in a human-in-the-loop setting.

5. **Access data from multiple file formats**: The community lacks consensus file formats for single-cell measurements and microscopy images. This can be attributed in part to the existence of proprietary file formats, an ecosystem fragmented along programming language boundaries and the nature of emerging experimental methods whose outputs do not conform to existing formats. While minimum information standards for metadata have been proposed[35,36], several file formats are commonly used to store single-cell and imaging data in the analysis ecosystem[32,34]. To address this challenge, Vitessce abstracts data loading from visualization rendering, enabling support for multiple file formats, including CSV, AnnData[20,32], MuData[21], SpatialData[9], Multivec[12], JSON, OME-TIFF[23] and OME-Zarr[24,25]. Data stored in novel file formats can also be visualized in Vitessce through the development of plugin data loading modules (Supplementary Note 1).

Vitessce enables visual exploration of single-cell experiments, which span the modalities of transcriptomics, epigenomics, proteomics and imaging within a single integrative tool. We demonstrate the construction, deployment and access of visualizations through use cases in Fig. 2 and Extended Data Figs. 1–7.

In a CITE sequencing use case, Vitessce can be used to validate the presence of cell types characterized by markers in both RNA and protein modalities. Stoeckius et al.[15] measure gene expression and surface protein abundance in a sample of human cord blood mononuclear cells, investigating well-characterized immune cell-type markers in these modalities. The findings of Stoeckius et al. can be reproduced by visualizing the dataset in Vitessce using linked scatterplots and heatmaps to explore protein abundance and gene expression

levels simultaneously (Extended Data Fig. 5). For instance, we observe that natural killer cells can be identified on the basis of CD56 protein levels and the expression of genes *GZMB*, *GZMK* and *PRF1*, as reported by the authors[15].

A second use case demonstrates visualization of spatially resolved gene expression data from a single-molecule fluorescence in situ hybridization experiment. In a mouse brain sample, Codeluppi et al.[37] discovered a transition region between the pyramidal L2/3 and L4 neuronal cell types that contains pyramidal L3/4 cells. This finding can be reproduced in Vitessce using the spatial plot and heatmap views (Fig. 2a). In Vitessce, the colors of cell segmentations and scatterplot points can be mapped to gene expression values. Through interactive selection of the genes *Lamp5* and *Rorb*, we observe the reported co-expression pattern that is specific to the pyramidal L3/4 cells[37].

The development of Vitessce has been propelled by collaboration with cell atlas consortia. Usage of the framework in the data portals for the Human BioMolecular Atlas Program (HuBMAP)[38] and Kidney Precision Medicine Project (KPMP)[39], as well as its open development on GitHub, has led to valuable feedback that has informed the requirements and future research directions. Future directions for Vitessce include comparative analysis support, increased scalability and additional visualization types for three-dimensional tissue maps and genome-mapped data.

Decoupled from any particular database, Vitessce can be integrated as a visualization frontend into projects with varying backend data organization strategies. The modularity and extensibility of the framework allows for supporting additional visualization types, file formats and single-cell modalities as they emerge. We anticipate that the community will embrace these characteristics by contributing plugin modules and incorporating Vitessce into applications that provide affordances for specific audiences, tasks and technologies.

## Online content

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

## Methods

### Using Vitessce

Vitessce is available in different forms to make it useful for multiple audiences. Researchers can use the R and Python packages to explore local or remote datasets during data analysis. The Vitessce website and online configuration editor are designed for sharing visualizations with collaborators and debugging. Software developers can incorporate Vitessce into other tools and write plugins by using it as a JavaScript package.

**Vitessce in Python environments and Jupyter Notebooks.** Vitessce can be used as a Python package in scripts and Jupyter notebooks. Visualizations can be configured and rendered directly into notebooks using an interactive widget that is compatible with multiple notebook environments including JupyterLab, Jupyter Notebook (classic) and Google Colab. The implementation of the widget is based on AnyWidget (https://anywidget.dev). Installation instructions and API documentation can be found at https://python-docs.vitessce.io. A set of tutorial notebooks is available at https://github.com/vitessce/vitessce-python-tutorial.

**Vitessce in R environments, RMarkdown, RStudio and Shiny apps.** Vitessce can be used as an R package in scripts, RMarkdown documents and Shiny apps. Visualizations configured in R can be rendered using an interactive widget in the RStudio Viewer pane, RMarkdown documents, pkgdown websites and Shiny apps. The widget is implemented as an R htmlwidget. Installation instructions and API documentation can be found at https://r-docs.vitessce.io.

**Vitessce in JavaScript and web environments.** Vitessce is implemented in JavaScript as a React component with corresponding APIs for configuration and registering plugins (described below). The JavaScript package can be used in websites, other React components or other JavaScript packages. Installation instructions and API documentation can be found at http://vitessce.io.

**Vitessce as a website and online configuration editor.** To quickly configure a Vitessce visualization from a web browser, we provide a web application to write and edit Vitessce configurations using JavaScript Object Notation (JSON) and JavaScript syntax. This method of configuring a Vitessce instance does not require Python, R or JavaScript package installation. This resource can be found at http://vitessce.io.

**Python, R and JavaScript configuration APIs.** Vitessce configurations are defined using a JSON representation that specifies the view layout and points to local or remote data files via URLs. In addition to the declarative JSON representation, we have developed Python, R and JavaScript APIs to enable users to define configurations programmatically. These configuration APIs support definition of datasets, files, views, the view layout and view coordinations within the native object-oriented paradigms of each language.

**JavaScript plugin API.** The Vitessce JavaScript package contains functions for defining plugin view types, coordination types, data types and file types. Once plugins have been defined, they must be registered by providing a name that will be used to refer to the implementation in Vitessce configurations. Plugin view types must be implemented as React components. We provide examples and tutorials for using the plugin API on the Vitessce documentation website.

### Multimodal configuration

Vitessce supports arbitrary multimodal datasets by adopting the observation-by-feature matrix conventions used by data analysis packages in the single-cell ecosystem including MultiAssayExperiment, Seurat and MuData. In this model, observations are entities being measured, such as cells, molecules, spots, beads or nuclei. Features are the characteristics being measured about entities such as genes, chromatin accessibility peaks or surface proteins. Feature values are the quantities being measured, such as expression levels, counts or intensities. Identifiers for types of observation, feature and feature value can be defined in the Vitessce configuration for both data and views. Vitessce then matches views to data accounting for observation type, feature type and/or value type identifiers. For most views and data types, all three properties are used, but a subset or a superset may be used depending on which properties are relevant to a particular view. For example, the heatmap considers all three properties when loading data because the visualization contains features, observations and values. A particular heatmap can be uniquely identified by these properties. In contrast, the feature list view is uniquely identified by only feature type and therefore observation type and value type are not considered when loading data for the view.

### Data organization

To separate data loading from rendering and support multimodal experiments, Vitessce views load data corresponding to datasets and data types, such as arrays of spatial coordinates per observation, dimensionality reduction coordinates per observation, images and observation-by-feature matrices. Views may load data corresponding to one or more datasets and one or more data types. These data types may be aligned on certain axes (for example, to support shared observation or feature sets) or not (for example, to support comparison of multiple datasets).

Data types are loaded independently such that their data may be contained in the same file or split across independent files, allowing multiple file formats to be used to load each dataset. Vitessce defines multiple file types that correspond to data type–file format pairs. If a file format supports multiple data types, a joint file type may be defined to simplify the configuration (that is, allowing URLs and file options to be defined once for a file while being exposed as multiple data types internally). For example, AnnData objects, which may contain multiple observation-by-feature matrices, dimensionality reductions and spatial coordinates, can be configured using a higher-level joint file type that is specific to AnnData. However these specifics are abstracted from the implementations of views that simply perform lookups for data corresponding to individual data types.

**On-demand loading of data subsets.** To scale to large datasets, data loading is deferred in multiple cases: multiscale images, multiscale bitmask-based segmentations, genome-mapped data and per-feature subsets of observation-by-feature matrices. The loading of highly multiplexed multiscale (that is, pyramidal) image files is implemented using Viv. As described by Manz et al.[11], Viv loads images as data tiles corresponding to the current viewport zoom level (that is, resolution) and target position (that is, $X/Y$), as well as selections for channel ($C$), temporal ($T$) and $Z$ axes. In Vitessce, this approach applies not only to primary images but also to image bitmask files for cell and organelle segmentations, which can also be stored in multiresolution formats.

Scalability to large observation-by-feature matrices in Vitessce takes advantage of a similar data tiling approach. These matrices can be stored in multiple Zarr-based file formats, in particular when using AnnData[19,32], MuData[21] and SpatialData[9]. Zarr supports tiled ('chunked') and compressed multidimensional arrays that can be served as directories of static files termed 'stores'. Benchmarking conducted by Moore et al.[24] demonstrates that accessing Zarr data is at least as fast as accessing the same data stored in hierarchical data format version 5 (HDF5) and TIFF, and in high-latency cloud storage scenarios, Zarr outperforms HDF5 by an order of magnitude.

The Zarr chunk strategy can be configured to optimize for particular use cases. In Vitessce, performance is optimized when many observations and few features are stored in each chunk, enabling use

cases such as quickly loading the expression values for a particular gene (that is, one feature) across all cells (that is, all observations). The trade-off is that the same chunk strategy might result in poor performance under a different use case such as loading expression values for all genes for only one cell. We additionally compare the full sizes of Zarr and CSV-based AnnData objects in Supplementary Fig. 7.

Genome browser tracks visualized using HiGlass also load data tiles on-demand based on the current browser viewport and zoom level, as described by Kerpedjiev et al.[12]. In Vitessce, we extend this mechanism to support genomic data from Zarr stores (in addition to the existing file formats supported by HiGlass) to eliminate the dependency on a specialized HiGlass Server. This extension is implemented as a HiGlass plugin data fetcher.

## Coordinated multiple views

Vitessce adopts the coordinated multiple views technique from the field of information visualization to enable comparison tasks, such as overview and detail, focus and context, difference views and small multiples[42,43].

**Coordination model.** The coordination model proposed by Boukhelifa et al.[29] is used to link subsets of views on visualization, interaction and data properties. In this model, views are not directly linked to one another, but instead to named property values referred to as coordination scopes. The properties that views can be linked on are referred to as coordination types. As a result, views are coordinated when they are linked to the same coordination scope for a given coordination type.

## JavaScript implementation

Vitessce is implemented as a configurable React component in Java-Script. Below the root React component, visualization and control views are also implemented as React components. Views for visualization may use any web technologies including WebGL, SVG, CSS and the HTML canvas element. The main Vitessce component is distributed in a JavaScript package that also exports an object-oriented configuration API and plugin APIs for defining custom views as React components, file types as JavaScript classes and coordination types.

**Implementation of coordinated multiple views.** We have implemented the aforementioned coordination model in JavaScript and incorporated it into the Vitessce configuration schema using a JSON representation. As we anticipate that this may be useful for general implementation of coordinated multiple view visualizations using JavaScript, we provide the implementation as a standalone JavaScript package that can be used in conjunction with the plugin API.

**Implementation of views.** Vitessce configurations define a set of views that contain interactive visualizations. View implementations are independent of one another and use custom React hook functions to access values from the coordination model and load data. The views currently available include a scatterplot, spatial plot, heatmap, cell set size bar chart, gene expression histogram, gene expression per cell set violin plots, cell set manager, gene selection, image layer controller and genomic profile per cell set genome browser. Views for visualization are implemented using custom JavaScript code and existing JavaScript libraries for statistical and geospatial data visualization. A list of the major open-source software packages used and a list of views currently available are provided in Supplementary Tables 1 and 2, respectively.

**Implementation of custom deck.gl layers.** The implementations of several views including the heatmap, scatterplot, and spatial views leverage the extensibility of the deck.gl API for WebGL-based data visualization. Deck.gl exposes not only high-level JavaScript APIs ('layers') for rendering points, lines, polygons and text, but also abstractions for developing custom layers with associated custom WebGL shader programs. The heatmap is implemented using a custom layer that performs aggregation on the graphic processing unit (GPU) of neighboring values in an observation-by-feature matrix when multiple values correspond to only a single pixel on the screen. This eliminates the aliasing and Moiré patterns that would otherwise occur at low zoom levels of a large matrix in the heatmap view, while preserving smooth zoom and pan interactions. The custom heatmap layer also contains logic which restricts pan interactions to the matrix area and determines how to display axis ticks on the basis of zoom level and text length. The spatial view is implemented using multiple custom layers, including one which renders image bitmasks by extending from Viv layers. The spatial and scatterplot views both use a custom layer, which efficiently maps feature values to colors on the GPU using quantitative colormap functions written in WebGL.

**Implementation of data loading.** Data are loaded using HTTP from static files on local or remote web servers. Files corresponding to one or more datasets are specified via an array of URLs and file types in the configuration. Certain file types accept options that specify details of the internal file organization to enable lookups (for example, to load particular arrays within Zarr stores by specifying their relative paths). Each file type is loaded by a corresponding JavaScript class that defines a required load function and optional functions to load data subsets (where file formats allow). Data loading classes may perform validation, in particular when file formats such as JSON are used, which may vary widely in their contents. A list of supported file types is provided in Supplementary Table 3.

## Processing of data for use cases

Data for use cases shown in Fig. 2 was processed using Python scripts, Jupyter notebooks and Snakemake pipelines[44] (Extended Data Figs. 1, 3 and 6). Vitessce visualizations were configured using the Vitessce Python package. Configurations were exported to JSON files and uploaded to the GitHub Gist service to enable referencing them by URL. Files obtained from the HuBMAP Data Portal were processed by automated pipelines developed within the HuBMAP Infrastructure and Engagement Collaboratory.

## Reporting summary

Further information on research design is available in the Nature Portfolio Reporting Summary linked to this article.

## Data Availability

All data shown in figures are available under permissive licenses. The smFISH dataset is available at http://linnarssonlab.org/osmFISH/ ref. 37. The Visium dataset is available from the Scanpy Python package (v1.9.3) using identifier 'V1_Human_Lymph_Node'. The CODEX dataset is available from the HuBMAP Data Portal using identifier 'HBM287.WDKX.539'. The CITE-seq dataset is available from NCBI Gene Expression Omnibus using identifier 'GSE100866'[15]. The Multiome dataset is available from the Mudatasets Python package (v0.0.2) using identifier 'brain3k_multiome', with the original dataset available via 10x Genomics at https://www.10xgenomics.com/resources/datasets/frozen-human-healthy-brain-tissue-3-k-1-standard-2-0-0. The Visium Spatial Gene Expression dataset from 10x Genomics was obtained through the Scanpy Datasets API[19], with the original dataset available at https://www.10xgenomics.com/resources/datasets/human-lymph-node-1-standard-1-1-0. The 3D multimodal mass spectrometry imaging dataset[40] is available via the HuBMAP Data Portal at https://portal.hubmapconsortium.org/preview/multimodal-mass-spectrometry-imaging-data. The CODEX dataset is available via the HuBMAP Data Portal at https://doi.org/10.35079/HBM543.RSRV.265 and https://portal.hubmapconsortium.org/browse/dataset/69d9c52bc9edb625b496cecb623ec081. The MALDI IMS (positive mode) dataset is available via the HuBMAP Data Portal at https://doi.org/10.35079/hbm876.xnrh.336 and https://portal.hubmapconsortium.org/browse/dataset/3ade70d66d10ed1d30fe005f672b2abf.

## Code availability

Vitessce is free open-source software available under an MIT license. Source code for the Vitessce JavaScript package is available via GitHub at https://github.com/vitessce/vitessce. The JavaScript package is distributed on NPM at https://www.npmjs.com/package/vitessce. Source code for the Python package containing configuration APIs and the Jupyter widget is available via GitHub at https://github.com/vitessce/vitessce-python. The Python package is distributed on PyPI at https://pypi.org/project/vitessce. The R package containing configuration APIs and the htmlwidget is available via GitHub at https://github.com/vitessce/vitessceR. Documentation and the online configuration editor are available at http://vitessce.io. JSON configurations used in figures are available via GitHub at https://gist.github.com/keller-mark/17417ec5c17228b3ca78b5edd3a7b89e. Code used to produce the visualizations that appear in figures of this manuscript is available via GitHub at https://github.com/vitessce/paper-figures. Code has been deposited in Zenodo at https://doi.org/10.5281/zenodo.11286222 (ref. 45), https://doi.org/10.5281/zenodo.11285945 (ref. 46), https://doi.org/10.5281/zenodo.11285962 (ref. 47) and https://doi.org/10.5281/zenodo.11285991 (ref. 48).

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

## Acknowledgements

Vitessce was developed with funding from the National Institutes of Health (OT2OD026677, N.G.; OT2OD033758, N.G.; T15LM007092, N.G.; T32HG002295, M.S.K.; R33CA263666, N.G.), the National Science Foundation (DGE1745303, T.M.), the Harvard Stem Cell Institute (CF-0014-17-03, N.G.) and the Chan Zuckerberg Initiative (CZIF2019-002450, P.V.K.). The use cases in this manuscript were supported with additional funding from the National Institutes of Health (U54DK120058, P.V.K.; 2P41GM103391, P.V.K.; OT2OD026671, P.V.K.) and National Science Foundation (CBET1828299, P.V.K.). We thank V. Petukhov for sharing the MERFISH mouse ileum dataset and assisting with its processing and interpretation. We thank J. Spraggins for the conceptualization of the imaging mass spectrometry algorithm comparison use case. We thank H. Tian, J. Tarolli, B. Stockwell and P. Rajbhandrari for the conceptualization of the three-dimensional multimodal imaging mass spectrometry use case. We thank M. Turner, S. L'Yi, E. Mörth, N. Akhmetov, J. Conroy, I. Babukova, T. Chan and J. K. Marx for feedback and other contributions to the project. Portions of Fig. 1 were created with BioRender.com.

## Author contributions

N.G. and P.V.K. conceived the project. N.G. coordinated the research team and oversaw the writing of the manuscript. M.S.K. led the design and implementation of the software with contributions from I.G., C.M. and T.M. C.M. implemented the first version of the software. All authors wrote and approved the final manuscript.

## Competing interests

N.G. is a co-founder and equity owner of Datavisyn. P.V.K. serves on the Scientific Advisory Board to Celsius Therapeutics, Inc. and Biomage, Inc. The other authors declare no competing interests.

## Additional information

**Extended data** is available for this paper at https://doi.org/10.1038/s41592-024-02436-x.

**Correspondence and requests for materials** should be addressed to Nils Gehlenborg.

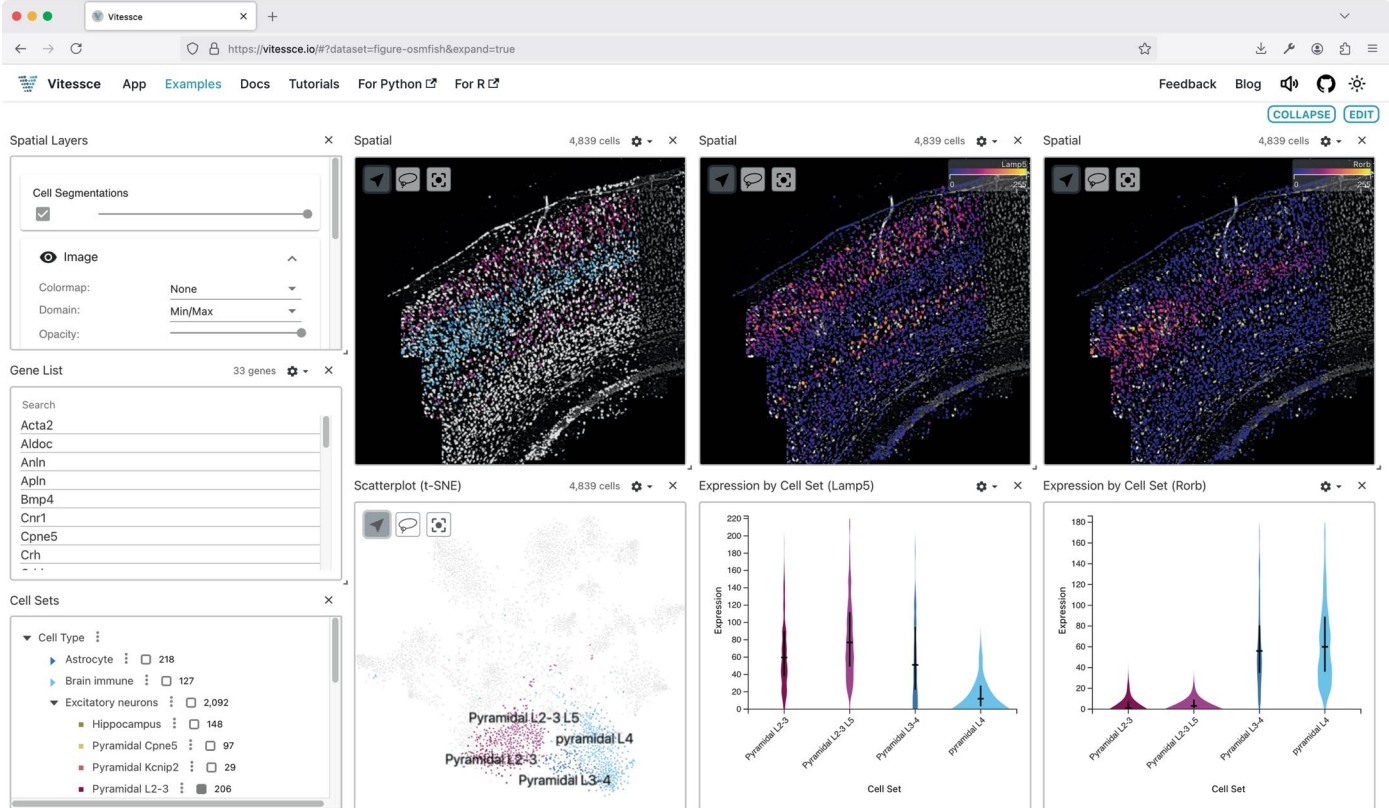

**Extended Data Fig. 1 | Visualization of single-molecule fluorescence in situ hybridization (smFISH) data.** Codeluppi et al.[37] used single-molecule fluorescence in situ hybridization (smFISH) to profile the somatosensory cortex in a mouse brain section. The authors selected 33 marker genes based on previous scRNA-seq findings in the somatosensory cortex and their ability to define cell types. A notable finding from this experiment was the discovery of a transition region defined by the Pyramidal L3/4 excitatory neuron cell type. Using the spatial plot and heatmap in Vitessce, we can reproduce this finding and observe the reported joint expression of markers *Lamp5* and *Rorb* that define the surrounding Pyramidal L2/3 and L4 cell types, respectively. URL: http://vitessce. io/#?dataset=figure-osmfish&expand=true Alternate URL: https://legacy. vitessce.io/demos/2024-07-26/24fdfb91/?dataset=figure-osmfish.

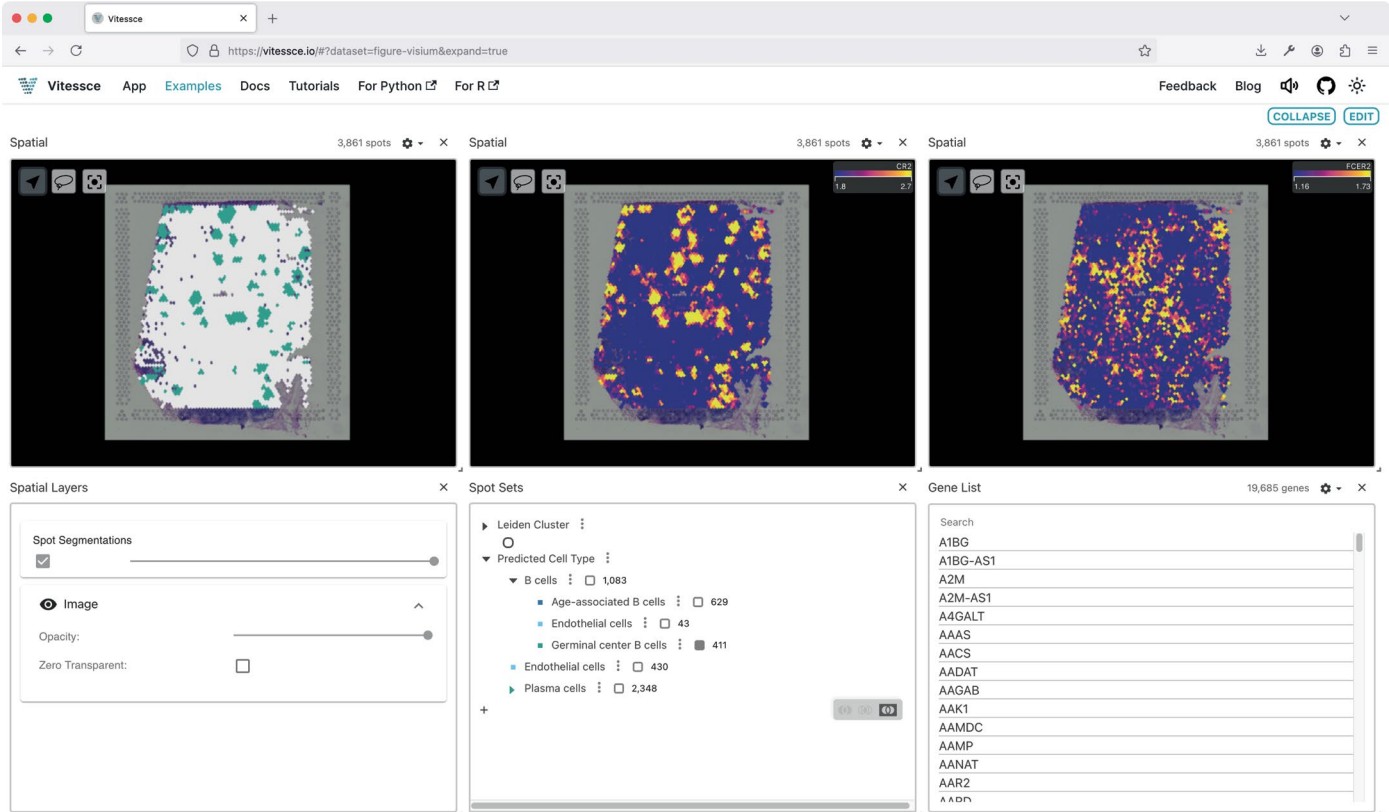

**Extended Data Fig. 2 | Visualization of a 10x Genomics Visium dataset.**
10x Genomics provides this dataset as a demo of the Visium technology and thus this dataset does not answer a particular biological question. Nonetheless, we can validate that the expected lymph node cell types are present. Using the CellTypist method for semi-automated cell type annotation of immune cells, we identify that *CR2* is highly expressed by spots predicted to contain Germinal center B cells[49]. Using CellPhoneDB[50], we query for known ligands of this receptor, which results in identification of *FCER2*[51]. Using linked spatial views in Vitessce, we can observe that *CR2* is specifically expressed by Germinal center B cell spots while the expression of *FCER2* is less specific but does overlap. URL: http://vitessce.io/#?dataset=figure-visium&expand=true Alternate URL: https://legacy.vitessce.io/demos/2024-07-26/24fdfb91/?dataset=figure-visium.

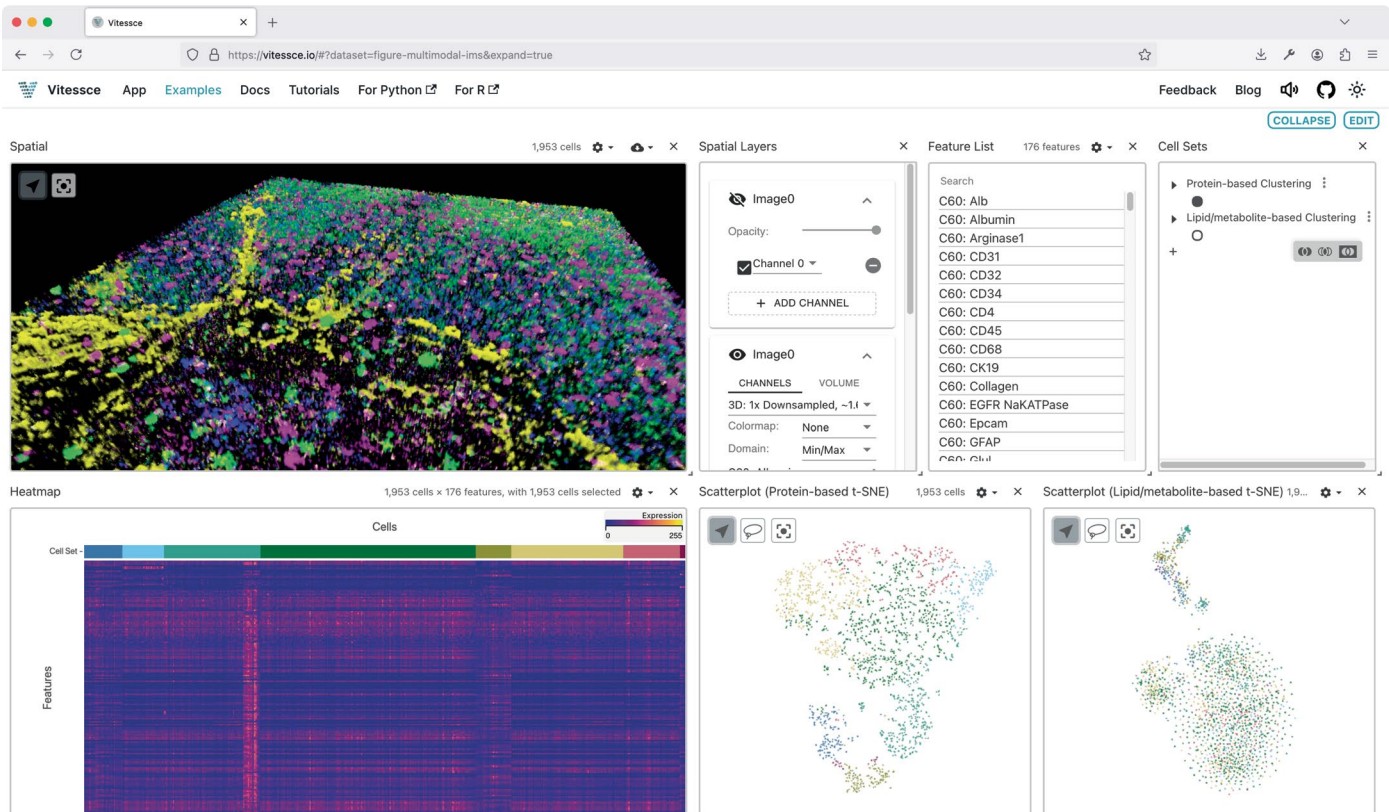

**Extended Data Fig. 3 | Visualization of a 3D multimodal mass spectrometry imaging dataset.** Tian et al.[40] report the development of a mass spectrometry imaging workflow that utilizes both cryogenic $(H_2O)_{n>28K}$-GCIB-SIMS and $C_{60}$-SIMS techniques to capture metabolites, lipids, and proteins at single-cell resolution in the same tissue section. Applying this workflow to a human liver sample, the authors find that they are able to classify metabolic zones and cell types using lipid and metabolite profiles. Using the volumetric rendering features available in Vitessce, we can explore the 3D multi-channel imaging data alongside the cell-by-feature heatmap and both protein-based and lipid/metabolite-based dimensionality reductions. URL: http://vitessce.io/#?dataset=figure-multimodal-ims&expand=true Alternate URL: https://legacy.vitessce.io/demos/2024-07-26/24fdfb91/?dataset=figure-multimodal-ims.

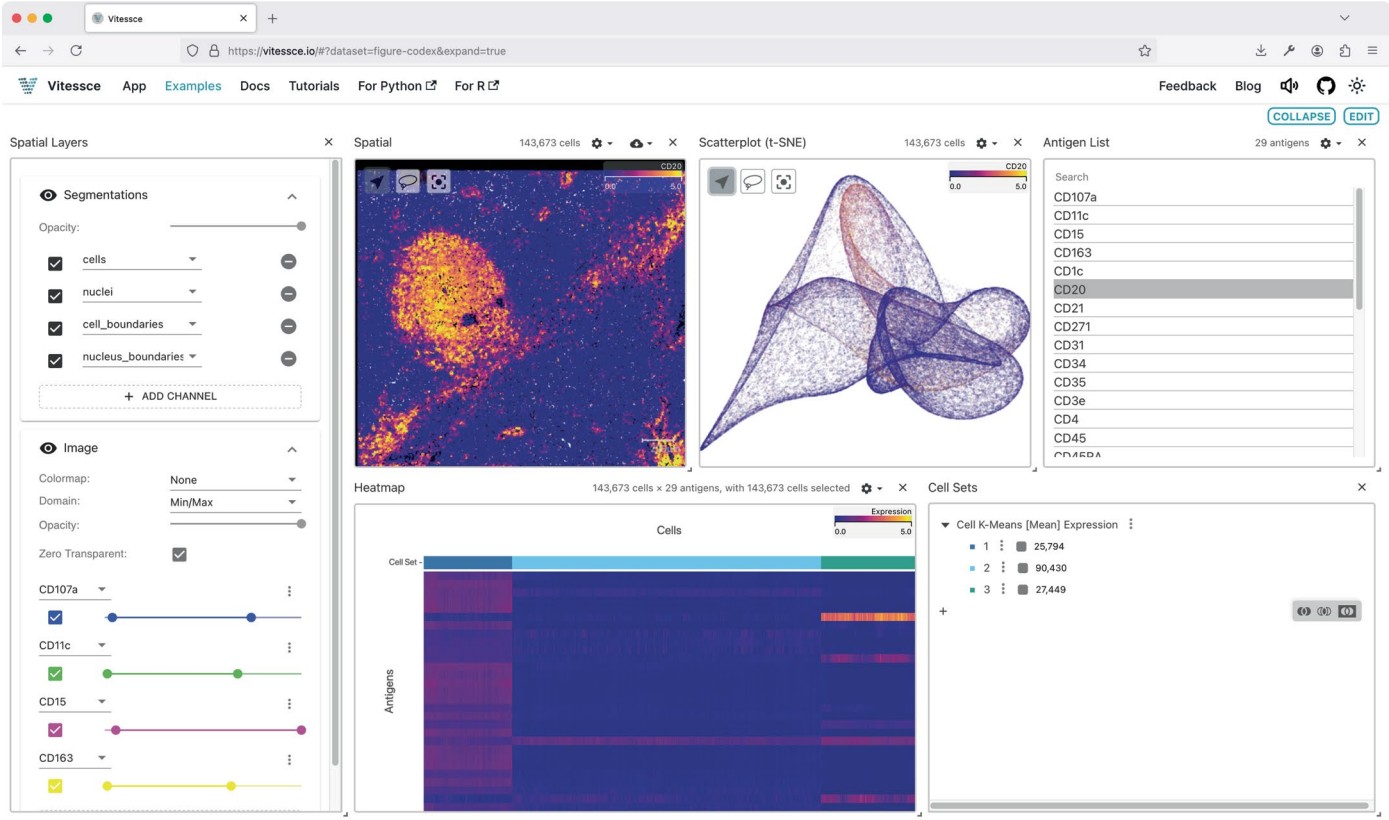

**Extended Data Fig. 4 | Visualization of a CODEX dataset.** CO-Detection by indEXing (CODEX) is a spatial assay which uses oligonucleotide probes conjugated to antibodies specific to a set of antigens of interest. Through multiple rounds of hybridization, up to 50 different antigens may be imaged. We present an example of using Vitessce to explore a human spleen tissue sample in which 29 antigens were targeted. The dataset contains the 29-channel microscopy images, cell segmentations, cell-by-antigen quantification, and unsupervised clustering results. To interact with this dataset, we configure Vitessce with a spatial plot, heatmap, and controllers to select image channels, antigens, and cell clusters of interest. With multiple representations of the same data, we can choose to begin exploration by focusing on one of several entity types. Because this dataset lacks cell type annotations, we are interested in the protein markers which define each cluster. We approach the visual analysis by selecting a cluster in the cell set controller, then searching for cluster-specific patterns in the spatial view. With clustering assignments encoded as cell segmentation mask colors, it appears that cells assigned to cluster 6 within the 'Cell K-Means [Mean-All-SubRegions] Expression' localize to a small set of compact regions in the spatial view, suggestive of cellular neighborhoods in the spleen. When we identify cluster 6 in the heatmap view, it is clear that the marker with the highest relative expression in this cluster corresponds to the B cell marker CD20. We can confirm that CD20 is enriched in this spatial region by hiding the cell segmentation masks in the spatial plot, uncovering the underlying image data. Using the spatial layer controller, we can select the CD20 channel and verify that the fluorescence signal indeed appears in the expected regions. The heatmap and image views can help us to find other markers correlated with the CD20 spatial expression pattern. URL: http://vitessce.io/#?dataset=figure-codex&expand=true Alternate URL: https://legacy.vitessce.io/demos/2024-07-26/24fdfb91/?dataset=figure-codex.

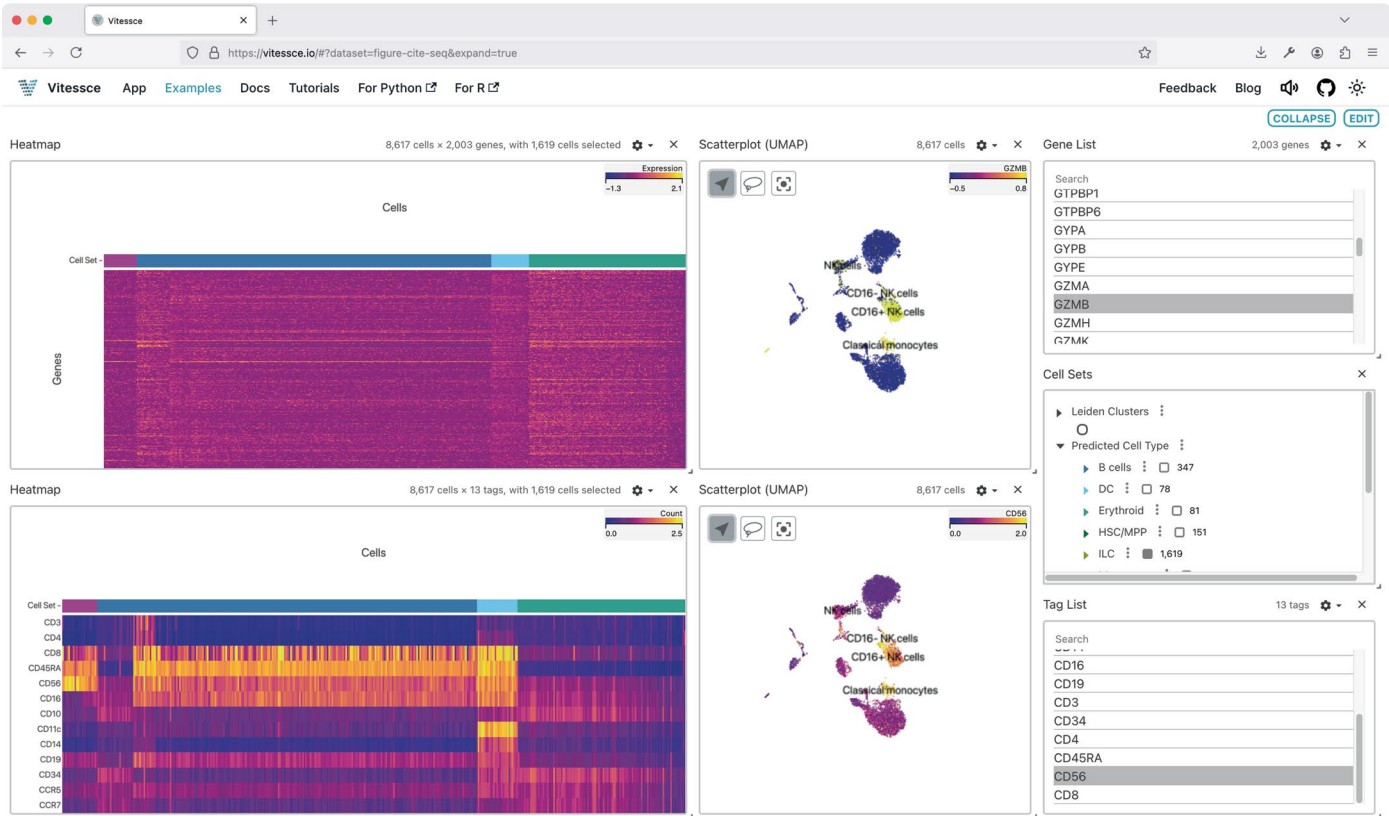

**Extended Data Fig. 5 | Visualization of a CITE-seq dataset.** Stoeckius et al. report the development of the CITE-seq method for measuring gene expression and surface protein abundance in the same cells[15]. The authors validate their technique on a sample of cord blood mononuclear cells (CBMCs) by measuring the abundance of well-characterized immune cell type markers. Using linked scatterplots and heatmaps to visualize protein abundance and gene expression levels simultaneously, we can reproduce the authors' multimodal characterization of the Natural Killer (NK) cell type based on CD56 levels and the expression of genes *GZMB*, *GZMK*, and *PRF1*. URL: http://vitessce. io/#?dataset=figure-cite-seq&expand=true Alternate URL: https://legacy. vitessce.io/demos/2024-07-26/24fdfb91/?dataset=figure-cite-seq.

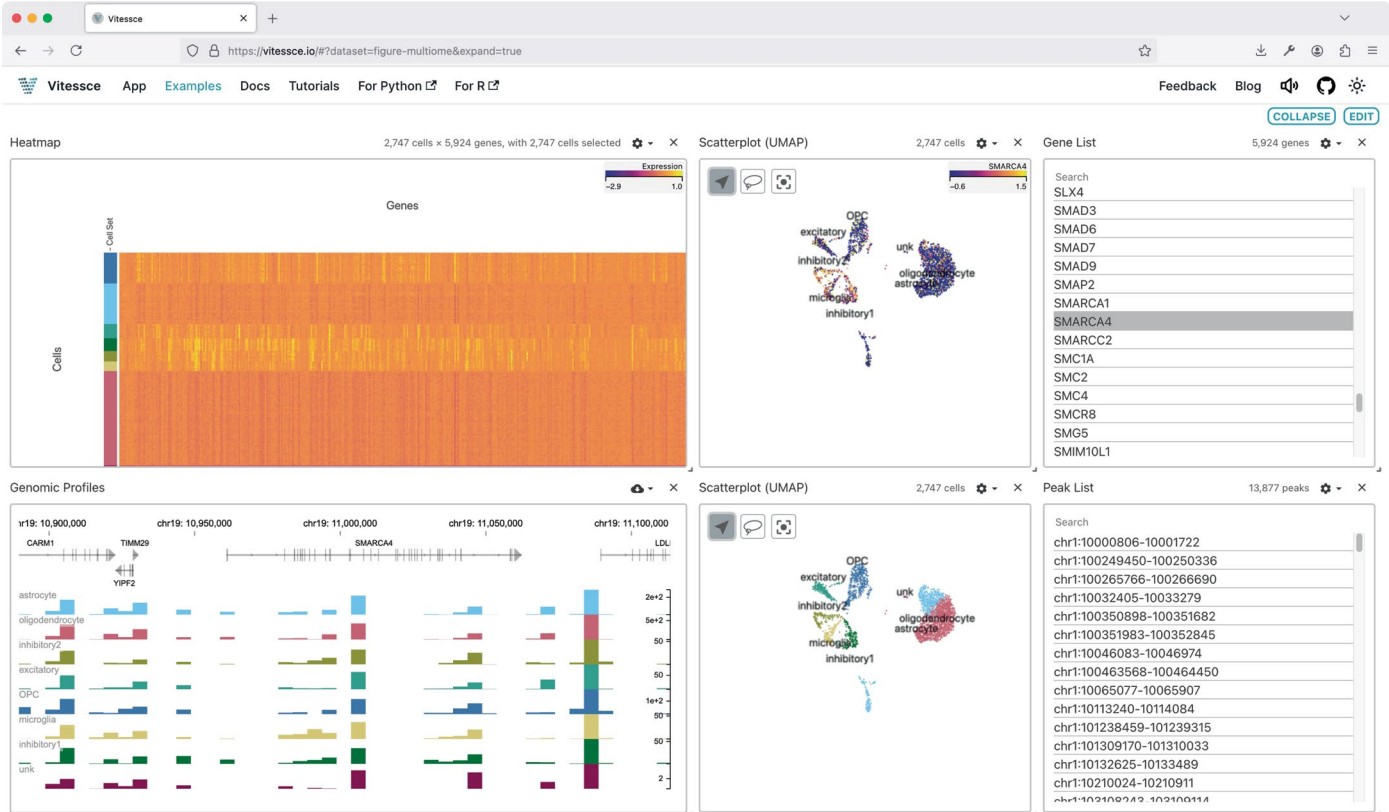

**Extended Data Fig. 6 | Visualization of a 10x Genomics Multiome dataset.**
This dataset is provided by 10x Genomics as a demo of the Multiome technology and thus is not intended to answer a particular biological question. Nonetheless, we can use this dataset to validate that the expected cerebellum cell types are present. Using the heatmap, we identify the gene *SYT1* based on its expression pattern in the cell cluster corresponding to the microglia cell type. Querying for transcription factors of *SYT1* using Cistrome Toolkit[52,53], we can identify *SMARCA4*

as the top result from a human neuron sample, with a regulatory potential score of 0.50982. Navigating to this gene on chromosome 19 in the genome browser view, we can observe a footprint-like pattern in the track for microglia, which we might want to validate in follow-up analyses. URL: http://vitessce. io/#?dataset=figure-multiome&expand=true Alternate URL: https://legacy. vitessce.io/demos/2024-07-26/24fdfb91/?dataset=figure-multiome.

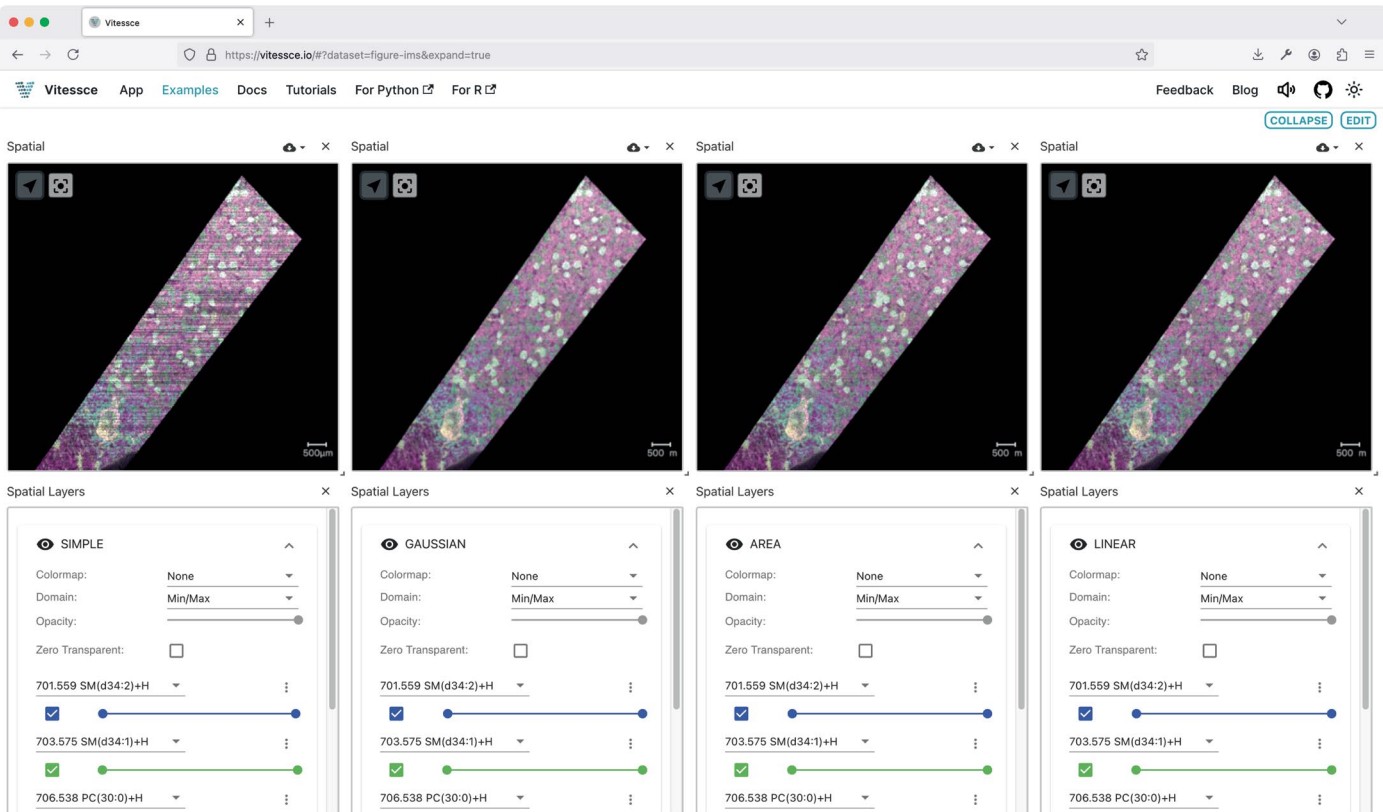

**Extended Data Fig. 7 | Visualization of a MALDI IMS dataset processed multiple ways.** Vitessce can be used for visual comparison of the outputs of multiple data processing methods within datasets containing multiple technical conditions, supporting the workflows of algorithm developers and computational biologists. For example, computational biologists tasked with analyzing imaging mass spectrometry (IMS) data may be interested in the consequences of using different interpolation methods for generation of pyramidal image files. Pyramidal image files contain downsampled lower resolution versions of the same image which are generated by pre-aggregating pixels, facilitating multi-scale visualization in a web browser. The choice of a downsampling pre-aggregation interpolation function may affect the visual properties of lower resolution images. The open microscopy image processing toolkit bioformats2raw implements several common interpolation methods:

SIMPLE, GAUSSIAN, AREA, LINEAR, CUBIC, and LANCZOS. We use the multi-scale image rendering capabilities of Vitessce to compare the application of these methods to an IMS dataset. The zoom levels and centers of each spatial plot are coordinated to facilitate comparison of the same image region at multiple scales. Vitessce helps to discover that the SIMPLE interpolation method introduces regularly-spaced horizontal streaks into the lowest resolution version of the image. Zooming in prompts Vitessce to load a higher resolution image in which the streaks are absent, indicating that they are artifacts of the downsampling procedure. This insight may steer us away from using the SIMPLE method when processing IMS data. URL: http://vitessce.io/#?dataset=figure-ims&expand=true Alternate URL: https://legacy.vitessce.io/demos/2024-07-26/24fdfb91/?dataset=figure-ims.

# Reporting Summary

## Statistics

For all statistical analyses, confirm that the following items are present in the figure legend, table legend, main text, or Methods section.

| n/a | Confirmed | |
|---|---|---|
| ☒ | ☐ | The exact sample size (*n*) for each experimental group/condition, given as a discrete number and unit of measurement |
| ☒ | ☐ | A statement on whether measurements were taken from distinct samples or whether the same sample was measured repeatedly |
| ☒ | ☐ | The statistical test(s) used AND whether they are one- or two-sided<br>*Only common tests should be described solely by name; describe more complex techniques in the Methods section.* |
| ☒ | ☐ | A description of all covariates tested |
| ☒ | ☐ | A description of any assumptions or corrections, such as tests of normality and adjustment for multiple comparisons |
| ☒ | ☐ | A full description of the statistical parameters including central tendency (e.g. means) or other basic estimates (e.g. regression coefficient) AND variation (e.g. standard deviation) or associated estimates of uncertainty (e.g. confidence intervals) |
| ☒ | ☐ | For null hypothesis testing, the test statistic (e.g. *F*, *t*, *r*) with confidence intervals, effect sizes, degrees of freedom and *P* value noted<br>*Give P values as exact values whenever suitable.* |
| ☒ | ☐ | For Bayesian analysis, information on the choice of priors and Markov chain Monte Carlo settings |
| ☒ | ☐ | For hierarchical and complex designs, identification of the appropriate level for tests and full reporting of outcomes |
| ☒ | ☐ | Estimates of effect sizes (e.g. Cohen's *d*, Pearson's *r*), indicating how they were calculated |

*Our web collection on statistics for biologists contains articles on many of the points above.*

## Software and code

Policy information about availability of computer code

| Data collection | No data was collected as part of this study. |
|---|---|
| Data analysis | Data analysis was performed using the Python programming language (v3.10) and the Python packages Numpy (v1.25.2), Pandas (v1.5.3), Scipy (v1.11.1), Jupyterlab (v4.0.3), Numba (v0.57.0), Scanpy (v1.9.3), Anndata (v0.8.0), Snakemake (v7.28.2), Zarr (v2.16.0), Tifffile (v2020.10.1), Scikit-image (v0.20.0), H5py (v3.9.0), Mudata (v0.2.3), Muon (v0.1.5), Celltypist (v1.5.3), Leidenalg (v0.9.1), Geopandas (v0.13.2), Loompy (v3.0.7), Generate-tiff-offsets (v0.1.7), Ome-zarr (v0.2.1), Negspy (v0.2.24), Pybbi (v0.3.5), Mudatasets (v0.0.2), Mofapy2 (v0.7.0), and Globus-cli (v3.16.0). |

For manuscripts utilizing custom algorithms or software that are central to the research but not yet described in published literature, software must be made available to editors and reviewers. We strongly encourage code deposition in a community repository (e.g. GitHub). See the Nature Portfolio guidelines for submitting code & software for further information.

## Data

Policy information about availability of data

All manuscripts must include a data availability statement. This statement should provide the following information, where applicable:

- Accession codes, unique identifiers, or web links for publicly available datasets
- A description of any restrictions on data availability
- For clinical datasets or third party data, please ensure that the statement adheres to our policy

All data shown in figures is available under permissive licenses. The smFISH dataset is available from http://linnarssonlab.org/osmFISH/. The Visium dataset is available from the Scanpy Python package (v1.9.3) using identifier 'V1_Human_Lymph_Node'. The CODEX dataset is available from the HuBMAP Data Portal using identifier 'HBM287.WDKX.539'. The CITE-seq dataset is available from NCBI Gene Expression Omnibus using identifier 'GSE100866'. The Multiome dataset is available from the Mudatasets Python package (v0.0.2) using identifier 'brain3k_multiome'.

## Human research participants

Policy information about studies involving human research participants and Sex and Gender in Research.

| | |
|---|---|
| Reporting on sex and gender | No information about sex or gender is described. |
| Population characteristics | This study does not involve human research participants. |
| Recruitment | This study does not involve human research participants. |
| Ethics oversight | This study does not involve human research participants. |

Note that full information on the approval of the study protocol must also be provided in the manuscript.

# Field-specific reporting

Please select the one below that is the best fit for your research. If you are not sure, read the appropriate sections before making your selection.

☒ Life sciences      ☐ Behavioural & social sciences      ☐ Ecological, evolutionary & environmental sciences

For a reference copy of the document with all sections, see nature.com/documents/nr-reporting-summary-flat.pdf

# Life sciences study design

All studies must disclose on these points even when the disclosure is negative.

| | |
|---|---|
| Sample size | No sample size calculations were performed because this paper does not report results of statistical analysis. |
| Data exclusions | No data were excluded from analysis. |
| Replication | No experimental findings or statistical analysis results are reported. |
| Randomization | No experimental findings or statistical analysis results are reported. |
| Blinding | No experimental findings or statistical analysis results are reported. |

# Reporting for specific materials, systems and methods

We require information from authors about some types of materials, experimental systems and methods used in many studies. Here, indicate whether each material, system or method listed is relevant to your study. If you are not sure if a list item applies to your research, read the appropriate section before selecting a response.

## Materials & experimental systems

| n/a | Involved in the study |
|-----|------------------------|
| ☒ | Antibodies |
| ☒ | Eukaryotic cell lines |
| ☒ | Palaeontology and archaeology |
| ☒ | Animals and other organisms |
| ☒ | Clinical data |
| ☒ | Dual use research of concern |

## Methods

| n/a | Involved in the study |
|-----|------------------------|
| ☒ | ChIP-seq |
| ☒ | Flow cytometry |
| ☒ | MRI-based neuroimaging |

