## [Peer Review File · Nature Methods]

Peer Review Information

Manuscript Title: Vitessce: integrative visualization of multimodal and spatially-resolved single-cell data

Corresponding author name(s): Professor Nils Gehlenborg

Editorial Notes: None

Reviewer Comments & Decisions:

Decision Letter, initial version:

11th Feb 2024

Dear Nils,

Thanks again for your patience.

Your Brief Communication, "Vitessce: integrative visualization of multimodal and spatially-resolved single-cell data", has now been seen by two reviewers. As you will see from their comments below, although the reviewers find your work of considerable potential interest, they have raised a number of concerns. We are interested in the possibility of publishing your paper in Nature Methods, but would like to consider your response to these concerns before we reach a final decision on publication.

We therefore invite you to revise your manuscript to address both the technical concerns and the concerns regarding presentation.

[REDACTED]

We hope to receive your revised paper within 2-3 months. If you cannot send it within this time, please let us know. In this event, we will still be happy to reconsider your paper at a later date so long as nothing similar has been accepted for publication at Nature Methods or published elsewhere.

OPEN SCIENCE REQUIREMENTS

REPORTING SUMMARY AND EDITORIAL POLICY CHECKLISTS

Please note that these forms are dynamic ‘smart pdfs’ and must therefore be downloaded and completed in Adobe Reader. We will then flatten them for ease of use by the reviewers. If you would like to reference the guidance text as you complete the template, please access these flattened versions at <http://www.nature.com/authors/policies/availability.html>.

DATA AVAILABILITY

All novel DNA and RNA sequencing data, protein sequences, genetic polymorphisms, linked genotype and phenotype data, gene expression data, macromolecular structures, and proteomics data must be deposited in a publicly accessible database, and accession codes and associated hyperlinks must be provided in the “Data Availability” section.

Please include a “Data availability” subsection in the Online Methods. This section should inform readers about the availability of the data used to support the conclusions of your study, including accession codes to public repositories, references to source data that may be published alongside the paper, unique identifiers such as URLs to data repository entries, or data set DOIs, and any other statement

about data availability. At a minimum, you should include the following statement: “The data that support the findings of this study are available from the corresponding author upon request”, describing which data is available upon request and mentioning any restrictions on availability. If DOIs are provided, please include these in the Reference list (authors, title, publisher (repository name), identifier, year). For more guidance on how to write this section please see: <http://www.nature.com/authors/policies/data/data-availability-statements-data-citations.pdf>

CODE AVAILABILITY

Please include a “Code Availability” subsection in the Online Methods which details how your custom code is made available. Only in rare cases (where code is not central to the main conclusions of the paper) is the statement “available upon request” allowed (and reasons should be specified).

MATERIALS AVAILABILITY

ORCID

Nature Methods is committed to improving transparency in authorship. As part of our efforts in this direction, we are now requesting that all authors identified as ‘corresponding author’ on published papers create and link their Open Researcher and Contributor Identifier (ORCID) with their account on the Manuscript Tracking System (MTS), prior to acceptance. This applies to primary research papers

only. ORCID helps the scientific community achieve unambiguous attribution of all scholarly contributions. You can create and link your ORCID from the home page of the MTS by clicking on 'Modify my Springer Nature account'. For more information please visit please visit www.springernature.com/orcid.

Sincerely,
Rita

Rita Strack, Ph.D.
Senior Editor
Nature Methods

Reviewers' Comments:

Reviewer #1:

Remarks to the Author:

The manuscript describes Vitesce, a framework for creating visualizations of single-cell omics and spatial omics in a variety of contexts and uses cases. Reading the paper, but in particular after accessing and trying the software, it is clear that Vitesce addresses a key requirement of the community. I note however, that the paper as written is not very compelling. Below I have some suggestions how to improve the presentation and clarify specific technical aspects of interest.

- Scope of the Vitesce. I would suggest strengthening the emphasis of Vitesce as a visualization framework. In particular, the list of unique features (which is a somewhat lengthy and tedious) does not sufficiently clarify that Vitesce is a framework to deploy flexible single-cell visualization solutions in a wide range of contexts and across compute infrastructures. This point should be strengthened.
- Multi-modal integration and preparation of input data. The description should clarify what is needed to prepare the data. In particular, do multi-modal data need to be preprocessed and integrated, e.g. using one of the many manifold inference methods? The nature of the processing within Vitesce versus

outside data preparation should be clarified. I would also suggest including information on filtering options, i.e. to select and apply quality metrics to filter data.

- Storage solutions and backend requirements for deployment. From the text it remains somewhat unclear what is needed to deploy Vitessce. Specifically, the needs to establish a Vitessce instance, both in terms of compute and IO should be defined. I assume that the IO requirements between the Vitessce instance and the data stores(s) could be substantial, specifically for large spatial omics modalities. I would also suggest highlighting some capabilities of lazy loading, multi-scale visualization, etc.
- Figures and presentation of use cases. The figures in the text are of poor quality and not particularly compelling. I specifically dislike the dump of screenshots in Figure 2. It might be an option to only include one screenshot and highlight another aspect of the framework.

Reviewer #2:

None

Reviewer #3:

Remarks to the Author:

Keller et al reports a new open source software tool, Vitessce, that enables visualisation of spatially-aware multi-modal data. A key contribution Vitessce is its versatility of enabling web-based data visualisation of a variety of data file types in popular programming languages in bioinformatics: R, Python. This feature is particularly helpful to enable Vitessce to be integrated to variety of bioinformatics pipelines.

This work is original and significant. The example codes are helpful. The paper is well written and clear.

However, I do have a few questions:

1. The author discussed using Zarr file type to overcome the scalability issues, however the author didn't not quantify the scalability results. Could the author provide more quantifiable results in the manuscript on the scalability testing, i.e. given X amount of cells how much time is taken to visualise that, and what is the maximal number of cells or file size that the website can accommodate?
2. Given that the App implementation is based on JavaScripts, the author didn't demonstrate how they did more low-level optimisation around the web technologies stack they used.
3. The demos in the Examples are helpful, but are limited to roughly 500 to 10K cells, could the author include demos that show performance of at least a million cells?
4. The App visualisation is in grey background for the cell clustering results, making it hard to see the data points, could this be changed?

Author Rebuttal to Initial comments

Reviewer 1

Remarks to the Author:

The manuscript describes Vitessce, a framework for creating visualizations of single-cell omics and spatial omics in a variety of contexts and use cases. Reading the paper, but in particular after accessing and trying the software, it is clear that Vitessce addresses a key requirement of the community. I note however, that the paper as written is not very compelling. Below I have some suggestions how to improve the presentation and clarify specific technical aspects of interest.

Thank you for the suggestions for improvement of the text and its organization. Below we detail how we addressed each point.

Comments

Scope of the Vitessce. I would suggest strengthening the emphasis of Vitessce as a visualization framework. In particular, the list of unique features (which is somewhat lengthy and tedious) does not sufficiently clarify that Vitessce is a framework to deploy flexible single-cell visualization solutions in a wide range of contexts and across compute infrastructures. This point should be strengthened.

We have strengthened the emphasis by stating in at the conclusion of the paragraph that introduces Vitessce: *"A strength of Vitessce that distinguishes it from prior work is the ability to deploy visualizations of single-cell data in a wide range of contexts and compute infrastructures, including in static websites, web applications, data portals, Jupyter Notebooks, RStudio, and R Shiny apps."*

Multi-modal integration and preparation of input data. The description should clarify what is needed to prepare the data. In particular, do multi-modal data need to be preprocessed and integrated, e.g. using one of the many manifold inference methods?

Thank you for these questions.

The answer to the question about multimodal data processing depends on whether the data analyst intends to visualize data post-integration or not.

For example, non-integrated data can be displayed in coordinated side-by-side visualizations (per-modality visualizations such as observation-by-feature heatmaps and dimensionality reduction scatterplots) which are linked based on shared observation IDs.

On the other hand, to display a single dimensionality reduction scatterplot that considers multiple modalities, an integration step would be required so that the dimensionality reduction coordinates reflect the features from the multiple modalities.

Finally, Vitessce can be used to visualize pre-integration and post-integration data simultaneously, for example with three linked scatterplots displaying the same cells from a SNARE-seq or 10x Multiome experiment (RNA-based UMAP, ATAC-based UMAP, and post-integration UMAP).

To address this in the text, we have added the following sentence: *“For multimodal datasets, analysts can use Vitessce to visualize modalities independently (i.e., different modalities in different plots) or to visualize the results of a data integration method that projects multiple modalities into a shared low-dimensional space.”*

The nature of the processing within Vitessce versus outside data preparation should be clarified.

To address this in the text, we have added the following sentence: *“Data preprocessing (e.g., dimensionality reduction and cell segmentation) must be performed outside of Vitessce or through plugins (Cheng et al. 2022), decoupling the visualizations from the bioinformatic algorithms.”*

I would also suggest including information on filtering options, i.e. to select and apply quality metrics to filter data.

Filtering based on quality metrics can be achieved via a preprocessing step prior to loading data into Vitessce, or via interactions with the visualizations in Vitessce.

Selections in Vitessce visualizations are based on sets of observations. Not only do we represent cell types and clustering results as sets of observations, but we also create sets of observations to store user-defined selections.

Vitessce supports loading quality metrics (e.g., percentage of mitochondrial transcripts per cell) as features and visualizing these quality metrics in statistical plots such as histograms. We demonstrate this in an example at <http://vitessce.io/#?dataset=habib-2017-with-quality-metrics&expand=true>. The user can select a quality metric in the bottom right, such as “percent_mito”. Then, the user can click and drag to select a range of values in the histogram plot. When finished, the cells with a corresponding percent_mito value will form a new user-defined selection of observations. This set of observations will appear in the set manager view and can be combined with other sets through set operations (union, intersection, and complement).

Another way for users to make selections is through interactive lasso of a region on the scatterplot or spatial views which results in the definition of a new user-defined set containing the selected observations. Then, users can interact with the set manager view to organize the

sets and operate on them using set operations. Statistical plots such as expression violin plots, histograms, and dot plots display only the currently-selected sets of observations, effectively enabling the user to filter by selection of sets.

In notebook environments in Python or R, after interacting with a Vitessce widget, users may access selected sets and observation IDs to use in downstream data processing code.

Vitessce also supports visualization of cell type annotation prediction scores using point opacity in the dimensionality reduction scatterplot. We demonstrate this feature in the example at <http://vitessce.io/#?url=https://gist.githubusercontent.com/keller-mark/a9ed636c6a993928c5c8f990b94c3032/raw/ca8fca59e3e714c95dd25e87fa5bfd7d7d89ac98/vitessce-obs-set-score-demo.json>. Prediction scores appear in the scatterplot (cells with lower scores appearing gray) and in tooltips (upon hovering cells in the scatterplot or heatmap).

Storage solutions and backend requirements for deployment. From the text it remains somewhat unclear what is needed to deploy Vitessce. Specifically, the needs to establish a Vitessce instance, both in terms of compute and IO should be defined. I assume that the IO requirements between the Vitessce instance and the data stores(s) could be substantial, specifically for large spatial omics modalities. I would also suggest highlighting some capabilities of lazy loading, multi-scale visualization, etc.

While there are multiple ways to use Vitessce such as using local Jupyter notebooks or using vitessce.io with a configuration, the recommended approach for long-term use cases (such as publications or data portals) would be for the visualization author to deploy a static website that embeds Vitessce as a component. To deploy an instance of Vitessce on a website that is publicly accessible on the internet, there are three steps required:

- Deploy the data to a static file hosting provider or cloud object storage system such as AWS S3 or Google Cloud Storage
- Define a Vitessce configuration that references the remote data files by their URLs.
- Deploy a static website that embeds Vitessce using a static web hosting provider.

Figures and presentation of use cases. The figures in the text are of poor quality and not particularly compelling. I specifically dislike the dump of screenshots in Figure 2. It might be an option to only include one screenshot and highlight another aspect of the framework.

Regarding the poor quality, we are able to provide a high-quality vector graphic version of Figure 1. We have reorganized the subfigures in Figure 1 and have added schematics representing the core view types available in Vitessce.

Thank you for the suggestion for improvement of Figure 2. We have redesigned this figure, reducing the number of screenshots (from 6 to 3). We have added diagrams that highlight the ability of the Spatial view to render multiple spatially-resolved data types using a layered approach (Figure 2D) and the ability to coordinate multiple view instances to achieve juxtaposition and superposition of visualizations (Figure 2E).

Reviewer 2

None

Reviewer 3

Remarks to the Author:

Keller et al reports a new open source software tool, Vitessce, that enables visualization of spatially-aware multi-modal data. A key contribution Vitessce is its versatility of enabling web-based data visualization of a variety of data file types in popular programming languages in bioinformatics: R, Python. This feature is particularly helpful to enable Vitessce to be integrated to variety of bioinformatics pipelines.

This work is original and significant. The example codes are helpful. The paper is well written and clear.

However, I do have a few questions:

Comments

1. The author discussed using Zarr file type to overcome the scalability issues, however the author didn't not quantify the scalability results. Could the author provide more quantifiable results in the manuscript on the scalability testing

Thank you for these questions.

While we did not perform extensive benchmarking of file formats as part of this study, we reference other studies that demonstrate the scalability of Zarr for data storage. Figure 1 of the publication by Moore et al. titled, "OME-NGFF: a next-generation file format for expanding bioimaging data-access strategies," compares the Zarr-based OME-NGFF image format to two existing formats, OME-TIFF and HDF5. They find the time to load imaging data from the Zarr-based format to be comparable among the three formats. To address this in the text, we have added the following sentence in the Data Organization subsection of Methods, "*Benchmarking conducted by Moore et al. demonstrates that accessing Zarr data is at least as fast as accessing the same data stored in HDF5 and OME-TIFF, and in high-latency cloud storage scenarios Zarr outperforms HDF5 by an order of magnitude.*"

Given that Zarr is a binary format (like OME-TIFF and HDF5), we would expect to observe performance improvements when compared to text-based formats for tabular data such as JSON and CSV.

Indeed, when we compare the file sizes for the same AnnData object saved to CSV and Zarr formats, we see that the Zarr-based files are significantly smaller than their CSV equivalents. The below figure compares sizes of CSV- and Zarr-based files used to store the same AnnData object with varying numbers of cells from 10 to 1,000,000. As suggested, we have included this in the manuscript in Supplementary Figure 14.

Further, the Zarr format allows for lazily loading subsets of data. For example, Vitessce requests only one column of the cell-by-gene expression matrix upon user selection of a gene. In contrast, the CSV format requires loading the full data matrix despite only being interested in a single column.

i.e. given X amount of cells how much time is taken to visualize that, and what is the maximal number of cells or file size that the website can accommodate?

We have created a new example with 1,283,972 cells at <http://vitessce.io/#?dataset=salcher-2022&expand=true> which takes less than one minute to load and render the visualization.

We acknowledge that web page load times are influenced by many factors including the data size, internet speed, location of data, device characteristics (e.g., memory,

CPU, and GPU specifications), web browser settings, and whether or not the content has been cached.

2. Given that the App implementation is based on JavaScript, the author didn't demonstrate how they did more low-level optimisation around the web technologies stack they used.

Vitessce uses multiple web technologies including WebGL for scalable visualization. We primarily use the DeckGL software (initially designed for large geospatial data visualization) which provides helpful abstractions on top of WebGL. We have also written custom WebGL shader code for usage by DeckGL, such as for rendering large multi-channel microscopy images, segmentation bitmasks, and the heatmap visualization. This is discussed in the manuscript in the Methods subsection titled, "*Implementation of custom DeckGL layers.*"

The microscopy image rendering aspects are described in the publication by Manz et al. that we reference titled, "Viv: multiscale visualization of high-resolution multiplexed bioimaging data on the web." Some of our team's code has been contributed back to the DeckGL open source library. We list usage of these libraries in Supplementary Table 1.

3. The demos in the Examples are helpful, but are limited to roughly 500 to 10K cells, could the author include demos that show performance of at least a million cells?

We have deployed an additional example demonstrating visualization of a dataset containing 1.2 million cells, available at <http://vitessce.io/#?dataset=salcher-2022&expand=true>

4. The App visualization is in grey background for the cell clustering results, making it hard to see the data points, could this be changed?

Thank you for this comment. Vitessce supports themes, including light and dark. We have added a new light theme with white background colors for all non-spatial visualizations, which is now the default light theme on the <http://vitessce.io> website. We have updated screenshots contained within figures to use this new theme.

Decision Letter, first revision:

5th Jul 2024

Dear Nils,

Thank you for submitting your revised manuscript "Vitesce: integrative visualization of multimodal and spatially-resolved single-cell data" (N METH-BC54413A). It has now been seen by the original referees and their comments are below. The reviewers find that the paper has improved in revision, and therefore we'll be happy in principle to publish it in Nature Methods, pending minor revisions to comply with our editorial and formatting guidelines.

TRANSPARENT PEER REVIEW

Please note: we allow redactions to authors' rebuttal and reviewer comments in the interest of confidentiality. If you are concerned about the release of confidential data, please let us know specifically what information you would like to have removed. Please note that we cannot incorporate redactions for any other reasons. Reviewer names will be published in the peer review files if the reviewer signed the comments to authors, or if reviewers explicitly agree to release their name. For more information, please refer to our FAQ page.

ORCID

Sincerely,
Rita

Rita Strack, Ph.D.
Senior Editor
Nature Methods

Reviewer #1 (Remarks to the Author):

Gehlenborg and colleagues have addressed my comments. I appreciate the revisions and in particular the new figures do look much better. I have no further comments.

Reviewer #1 (Remarks on code availability):

I have looked through a number of vignettes and we are already starting to use Vitessce in some of our own projects. I am confident that the software is sound and sufficiently mature for publication.

Reviewer #3 (Remarks to the Author):

The authors have addressed my main concerns about scalability of this tool. In particular, the new example of 1.2M cells was helpful to demonstrate its usability to a suitably large single cell data set. The revised text has clarified a number of issues raised by myself and the other reviewer. I believe this tool is useful for the single cell analysis community.

Reviewer #3 (Remarks on code availability):

The documentation of various versions of the code (R, Python and JavaScript) was sufficient and helpful.

Final Decision Letter:

Dear Nils,

I am pleased to inform you that your Brief Communication, "Vitesce: integrative visualization of multimodal and spatially-resolved single-cell data", has now been accepted for publication in Nature Methods. The received and accepted dates will be Nov 9, 2023 and August 16, 2024. This note is intended to let you know what to expect from us over the next month or so, and to let you know where to address any further questions.

Over the next few weeks, your paper will be copyedited to ensure that it conforms to Nature Methods style. Once your paper is typeset, you will receive an email with a link to choose the appropriate publishing options for your paper and our Author Services team will be in touch regarding any additional information that may be required.

Once proofs are generated, they will be sent to you electronically and you will be asked to send a corrected version within 48 hours. It is extremely important that you let us know now whether you will be difficult to contact over the next month. If this is the case, we ask that you send us the contact information (email, phone and fax) of someone who will be able to check the proofs and deal with any last-minute problems.

If, when you receive your proof, you cannot meet the deadline, please inform us at rjsproduction@springernature.com immediately.

Please note that *Nature Methods* is a Transformative Journal (TJ). Authors may publish their research with us through the traditional subscription access route or make their paper immediately open access through payment of an article-processing charge (APC). Authors will not be required to make a final decision about access to their article until it has been accepted. Find out more about Transformative Journals

Authors may need to take specific actions to achieve compliance with funder and institutional open access mandates. If your research is supported by a funder that requires

immediate open access (e.g. according to Plan S principles) then you should select the gold OA route, and we will direct you to the compliant route where possible. For authors selecting the subscription publication route, the journal's standard licensing terms will need to be accepted, including self-archiving policies. Those licensing terms will supersede any other terms that the author or any third party may assert apply to any version of the manuscript.

If you are active on Twitter/X, please e-mail me your and your coauthors' handles so that we may tag you when the paper is published.

To assist our authors in disseminating their research to the broader community, our SharedIt initiative provides you with a unique shareable link that will allow anyone (with or without a subscription) to read the published article. Recipients of the link with a subscription will also be able to download and print the PDF. As soon as your article is published, you will receive an automated email with your shareable link.

Please note that you and your coauthors may order reprints and single copies of the issue containing your article through Springer Nature Limited's reprint website, which is located at <http://www.nature.com/reprints/author-reprints.html>. If there are any questions about reprints please send an email to author-reprints@nature.com and someone will assist you.

Best regards,
Rita

Rita Strack, Ph.D.
Senior Editor
Nature Methods